# Nanoarchitectonics for Biodegradable Superabsorbent Based on Carboxymethyl Starch and Chitosan Cross-Linked with Vanillin

**DOI:** 10.3390/ijms23105386

**Published:** 2022-05-11

**Authors:** Elżbieta Czarnecka, Jacek Nowaczyk, Mirosława Prochoń, Anna Masek

**Affiliations:** 1Physical Chemistry and Physico-Chemistry of Polymers, Faculty of Chemistry, Nicolaus Copernicus University in Toruń, 7 Gagarina Street, 87-100 Toruń, Poland; 2Plastica Sp. Z O.O., Frydrychowo 55, 87-410 Kowalewo Pomorskie, Poland; 3Institute of Polymers and Dye Technology, Faculty of Chemistry, Lodz University of Technology, 16 Stefanowskiego Street, 90-537 Lodz, Poland; miroslawa.prochon@p.lodz.pl (M.P.); anna.masek@p.lodz.pl (A.M.)

**Keywords:** carboxymethyl starch, chitosan, biodegradation, hydrogel, superabsorbent polymer, polysaccharide, biomaterial

## Abstract

Due to the growing demand for sustainable hygiene products (that will exhibit biodegradability and compostability properties), the challenge of developing a superabsorbent polymer that absorbs significant amounts of liquid has been raised so that it can be used in the hygiene sector in the future. The work covers the study of the swelling and dehydration kinetics of hydrogels formed by grafting polymerization of carboxymethyl starch (CMS) and chitosan (Ch). Vanillin (Van) was used as the crosslinking agent. The swelling and dehydration kinetics of the polymers were measured in various solutes including deionized water buffers with pH from 1 to 12 and in aqueous solutions of sodium chloride at 298 and 311 K. The surface morphology and texture properties of the analyzed hydrogels were observed by scanning electron microscopy (SEM). The influence of this structure on swelling and dehydration is discussed. Fourier transform infrared (FTIR) analyses confirmed the interaction between the carboxymethyl starch carbonyl groups and the chitosan amino groups in the resulting hydrogels. Additionally, spectroscopic analyses confirmed the formation of acetal crosslink bridges including vanillin molecules. The chemical dynamics studies revealed that new hydrogel dehydration kinetics strongly depend on the vanillin content. The main significance of the study concerns the positive results of the survey for the new superabsorbent polymer material, coupling high fluid absorbance with biodegradability. The studies on biodegradability indicated that resulting materials show good environmental degradability characteristics and can be considered true biodegradable superabsorbent polymers.

## 1. Introduction

Superabsorbent polymers (SAPs) found their large-scale industrial application in the mid-1970s as an active additive to the absorbent core of hygiene products. Since then, the development of the hygiene products industry was coupled with the evolution of superabsorbent materials. The term superabsorbent is attributed to substances capable of absorbing fluids in amounts exceeding 100 times their dry mass [1]. The ability to absorb large amounts of liquid and bind it inside the polymer network is a common feature of most polymeric hydrogels, making these polymers an obvious source of SAP candidates. The most efficient SAPs were those with a polyacrylate polymeric backbone. They are now widely used and studied as hydrogel polyelectrolytes owning their properties to the presence of ion genic side groups. The solvation process (usually hydration) of ions fixed to a polymer network results in the enormous swelling of the material. The solvent trapped in the solvation spheres of those ions is bonded strong enough to prevent leakage from the swollen material.

The application of SAPs as active additives to the absorbent core of the hygienic product allowed the design of goods, such as superabsorbent disposable diapers and hyper-thin hygienic pads, which became important parts of modern life in developed countries. At the beginning of the SAPs evolution, two factors played the leading role in controlling the direction of further innovations. They were, increased sorption capacity and sorption stability. Recently, however, those factors have been overwhelmed by another related to the increasing environmental awareness of consumers, the biodegradability of postconsumer waste. Consequently, the important task driving the innovations in the hygiene products industry is the development of biodegradable SAPs [2]. It seems an irony because following the history of SAP development one can notice that the first patent on the use of SAPs as an absorbent in diapers concerned chemically crosslinked starch and cellulose, i.e., biodegradable polymers [3].

The trends of contemporary society to buy eco-friendly goods together with depleting crude oil resources, motivate scientists and research centers to increase their interest in biopolymer-based materials [4,5,6]. As part of a large group of natural polymers, starch plays an important role due to its positive characteristics: non-toxicity, biocompatibility, renewable, easy availability, and low price, which makes it widely used in many industries [7,8]. Native starch suffers; however, from several drawbacks, such as difficulty to control viscosity after gelation, the tendency to retrogradation, insolubility in cold water, clouding of gels/water solutions, unsatisfactory mechanical properties, and rapid degradation. In order to minimize the negative properties of native starch, a variety of chemical modifications are increasingly being used [9].

According to the literature, carboxymethyl starch (CMS) is one of the most important starch derivatives. The first scientific report on CMS was published in 1924 by Chowdhury and has been of great interest ever since [10]. Carboxymethylation of cassava, corn, and potato starch already belonged to the most common methods of CMS synthesis. The process involves the etherification of free hydroxyl groups of starch with carboxymethyl groups (–CH_2_–COOH) [11]. Carboxymethyl starch is a water-soluble derivative with an ion genic group bearing a negative charge after dissociation. It can be treated as a green polymer that is useful in many industries, such as environmental protection, cosmetics, medicine, pharmacy, food industry, and many others [12]. The degree of substitution of glucoside units determines the properties of the synthesized CMS, such as pH, gelatinization temperature, the viscosity of aqueous solutions, stability during storage, and dissolution rate [13,14,15].

Chitosan is a linear polysaccharide that is commonly available and partially acetylated (1-4)-2-amino-2-deoxy-β-glucan [16,17]. It is obtained from mushrooms, shrimp shells, and crustaceans and commercially produced by the deacetylation of chitin [18] with various degrees of deacetylation (DDA) and molecular weight (Mw). Chitosan is a weak base polyelectrolyte, biocompatible, biodegradable, and biofunctional; it is insoluble in water and common organic solvents. On the other hand, it easily dissolves in aqueous solutions of organic acids at a pH below 6.3, due to the conversion of glucosamine units into a protonated form NH_3_^+^ [19]. Solvation of these cations results in significant chain separation and a decrease in the intermolecular forces preventing dissolution in water.

The evolution of SAPs focused on mastering their performances lead through the introduction of acrylic monomers bearing ion genic groups. This includes the introduction of crosslinkers with multiple vinyl moieties. Recently, increasing attention has been paid to green and natural chemistry and so scientists are searching for natural cross-linking factors that will eliminate toxic and nondegradable vinyl compounds. In this study, vanillin was used as a cross-linking agent due to its aldehyde group. Vanillin (4-hydroxy-3-methoxybenzaldehyde) produced from sugar beet and vanilla pods has many applications in the pharmaceutical, perfumery, beverage, and food industries. The aim of the study was to obtain a biodegradable superabsorbent derived only from natural substrates. The obtained products with different content of cross-linking agents were analyzed by thermogravimetric analysis (TGA), scanning electron microscopy (SEM), and infrared Fourier spectroscopy (FTIR); the swelling properties in the water, sodium chloride solutions, and various aqueous solutions with different pH were examined.

## 2. Results and Discussion

### 2.1. Chemical Structure of the Products

Due to the specific properties of the polymers obtained in the course of the study, particularly the insolubility in common solvents necessary for most analytical methods, the only reasonable way to investigate their chemical structure was infrared spectroscopy. However, this is not the best way for chemical structure determination; it is widely used and generally accepted as a common way of insoluble polymeric materials’ characterization [20] The analysis of reaction mechanisms and the possible structure of products was based on the general knowledge of reagent properties and similar reports in the literature [21].

Both chitosan and CMS are members of a polysaccharide family characterized by repeating units derived from glucopyranose rings. The repeating units of chitosan differ from basic glucopyranose in one amine group at position 2 and CMS in the carboxymethylated OH group at C6. Apart from the above-mentioned features, the structures are similar, with the presence of two hydroxyl groups susceptible to the formation of hydrogen bonds. The initial polymers are water-soluble and form a compatible polymer blend. According to the literature, chitosan blends are mainly prepared through solution mixing of the polymers [22]. Chitosan amine groups present on the main chain introduce specific reactivity to the polymer. In practice, they mainly serve as hydrogen bond donors or acceptors. These amine groups can also serve as Lewis bases, controlling the polymer dissolution dependence on pH and taking part in specific reactions, i.e., with carbonyl groups. In the aqueous environment, the competition with abundant water molecules decreases the likelihood of reaction between –NH_2_ and >C=O but this kind of interchain bonding between CMS and chitosan results in the formation of a 3D polymer network regarded as an interpenetrating network. In order to verify the formation of amide linkages between chitosan’s amine group and CMS’s carbonyl substituent, the analysis of adequate infrared spectra was conducted according to the following procedure. First, the spectra of initial polymers and chitosan/CMS mixture were recorded and normalized with respect to a band at 2912 cm^−1^. The band corresponds with antisymmetric C–H stretching that was found to be invariant in the studied systems. After normalization, the spectra of initial polymers were numerically added using software delivered with the apparatus. Then the simulated spectrum was overlapped with the corresponding spectrum of polymer mixture and their juxtaposition is presented in Figure 1. As shown in the graph, the simulated spectrum (blue line) is similar to the real spectrum (black line) of the mixture. Apart from some intensity fluctuation that can be attributed to the differences in data manipulation necessary to obtain simulated spectra, a few differences can be pointed out as significant.

In the spectrum of reagents (blue), there are visible bands at 1664, 1590, and 1510 cm^−1^, absent in the spectrum of the mixture (black) in this region, though, a broadband with a maximum at 1555 cm^−1^ can be found. The explanation of these changes seems straightforward since the bands at 1664 and 1590 cm^−1^ can be attributed to SMS’s carboxyl group and the bands at 1510 cm^−1^ correspond with the primary amine group in chitosan, their disappearance indicates the reaction involving these groups. Such a reaction results in the formation of an amide junction. The spectroscopic proof of amides is the occurrence of the band with a maximum at 1555 cm^−1^ in the spectra of the mixture and a small new signal at 1321 cm^−1^. Both the bands correspond with amide group vibrations. To achieve conditions convenient for this kind of crosslinking the polymer coils need to be untangled and separate counterpart chains should occupy neighboring space. Observed spectroscopic features prove that the situation occurs frequently enough to be noticed on the FTIR spectra.

To promote the crosslinking necessary for an appropriate hydrogel that can be further dried to obtain a superabsorbent gel, a crosslinking agent was added. In our case, the role was given to vanillin having a reactive carbonyl group attached to the phenyl ring. The low-molecular-weight molecules of vanillin can easily penetrate the interior of swollen coils of the polymers and interact with amine and/or hydroxyl groups. The aldehyde group of vanillin can react with amine groups on chitosan chains forming imine links –N=CH– (see Figure 15) [21,23] the loose end of the vanillin residue has a hydroxyl group serving as a hydrogen bond donor or acceptor. This hydroxyl group can form strong hydrogen bonds with hydroxyl groups of glucose residues on other chains or other parts of the same chain, preferably with the –OH group attached to C-6 since it has less spherical hindrances than the other hydroxyl groups in a chitosan repeatable unit. This leads to crosslinking between chitosan chains. Alternatively, the end could be blocked by a hydrogen bond with acetic acid present in the mixture. Another possible crosslinking of chitosan by vanillin according to literature involves the reaction of the aldehyde group from vanillin with a hydroxyl group on the chain-forming hemiacetal, a further reaction with another hydroxyl group affords an acetal bridge between the two chains. The mechanism of this reaction is given in the literature [21]. Although this kind of reaction is not discussed in the literature as a typical chemical modification of polysaccharides [23], the spectroscopic analyses seem to support the occurrence of this reaction. Acetal bridges are characterized by a specific band at about 1005 cm^−1^ and in the case of polysaccharides it will overlap with the variety of C–O–C bonds present in the polymer chain. Nevertheless, a thorough analysis of FTIR spectra recorded for the Chitosan/CMS/Vanillin system containing increasing content of the last has shown a significant increase of the band centered at 1005 cm^−1^.

The investigation of this mechanism is based on the infrared study of vanillin content influence on the polymer spectrum. In order to study the effect of a set of polymers, Chitosan/CMS/Van with different content of vanillin was prepared and studied spectroscopically. The recorded spectra of these polymers were qualitatively similar as shown in the original spectra included in Appendix A. The spectra were transformed into absorption curves and normalized with respect to a band at 2912 cm^−1^. The intensity of the most prominent bands was tabularized and subjected to correlation analysis. The collection of specific intensities is shown in Table 1.

Based on the specific peak intensity the correlation analysis was conducted and the corresponding correlation matrix is given in Table 2.

It was found that the highest correlation with vanillin content was with bands at 1064 and 1005 cm^−1^, which are correlated with each other since they correspond with the vibration of the same family of bonds C–O–C. There is also a significant correlation with the band at 3011 cm^−1^ corresponding with the C–H stretching vibration of the benzene ring, which is characteristic of vanillin as the only aromatic compound in the system. Figure 2 shows the plots of respective relationships.

The intensity of bands corresponding with aromatic ring C–H vibrations are small and partially disturbed by the noise which explains the not ideal correlation. On the other hand, the strongest band in the spectra at 1005 cm^−1^ shows logarithmic dependence of C–O–C absorbance and crosslinker concentration. According to this, one can conclude that although the acetal crosslinking increases with the vanillin content, at some point it will reach its maximum and further addition of crosslinker would not affect the polymer.

Apart from acetal links, vanillin forms also amide bonds with the amine groups of chitosan which is evidenced by the significant correlation between vanillin content and amide bands (especially the so-called amide III band at 1375 cm^−1^).

It is important to remember that hydrogen bonds play important role in such systems and these bonds are relatively stable at low temperatures and can be easily broken by small polar molecules, e.g., acids and ethanol. Too many hydrogen bonds in the system lead to a reduction of the elasticity of the polymer. Therefore, small molecules of vanillin competing in hydrogen bond formation with neighboring chains can act as plasticizers. Chitosan has a “rigid” structure of chains, which hindered the diffusion of liquids; the use of vanillin and its combination with carboxymethyl starch caused the chains to relax. Appropriate free mobility of polymer molecules and an appropriate amount of free amino groups contribute to increasing the absorptive properties of this hydrogel.

### 2.2. Infrared Spectra Discussion

The FTIR technique was used to identify the structural features of obtained materials (Figure 3 and Appendix A). 

In the FTIR spectrum of carboxymethyl starch, there is a broad peak between 3000 and 3500 cm^−1^, which corresponds to the O-H stretch vibration. Another band at about 2927 cm^−1^ can be attributed to the stretching vibrations of C–H bonds [9,24]. The carboxyl group in the CMS shows intense absorption peaks at 1589 cm^−1^ and 1411 cm^−1^, resulting from symmetric and asymmetric vibrations, respectively [25,26]. The spectrum of chitosan shows a broad absorption band ranging from 3000 to 3500 cm^−1^, which results from overlapping signals of stretching vibrations of N–H and O–H bonds in –NH_2_ and –OH groups, respectively. Similar to the previous case, the two bands at 2926 cm^−1^ and 2873 cm^−1^ correspond to the stretching modes of C–H bonds [27]. The bands located at the 1665 and 1638 cm^−1^ correspond to the stretching vibration of the C=O bonds of the acetylated units (so-called amide I vibrations of N–C=O), usually reported in the range of 1649−1667 cm^−1^. The peak at about 1588 cm^−1^ can be assigned to the antisymmetric deformations of the amine –NH_2_ group (the same band is suspected to occur in protonated primary amines) [28] characteristic of non-acylated units [9]. The skeletal vibrations including the C–O stretching typical for saccharide structures occur at 1022 and 1062 cm^−1^ [28,29]. Synthesized polymers differ in the content of natural counterparts (chitosan and carboxymethyl starch) and the amount of biological crosslinker (vanillin). 

### 2.3. Thermal Analysis

When planning to obtain excellent hygiene products, the thermal properties of the materials should be taken into account. This is important due to the various conditions of storage and use, and most importantly, when the product comes into contact with the body of an adult or a child. The thermal analysis method was used to investigate the thermal stability and degradation profile of simple polymers (chitosan and carboxymethyl starch) and their superabsorbent polymers cross-linked with vanillin. Numerical values of the tests are collected in Table 3.

The five percent weight loss for all samples at about 120 °C represents the amount adsorbed by hydroxyl and amine groups and bound water. When analyzing the thermal decomposition of CMS, it can be seen that the main decomposition stage occurs at about 263 °C with about a 35% weight loss. The degree of substitution (DS) of acetylated starches influences their thermal stability. The higher the DS, the lower the thermal stability due to the hydrophilic nature of the carboxymethyl groups which facilitate thermal decomposition [30]. The TGA curve of pure chitosan shows the main degradation step at about 265 °C with a weight loss of about 44%, which may be related to chain breakdown depolymerization and cleavage of glycosidic bonds [31]. The TGA/DTG curves of raw materials and obtained polymers are rather similar. The main stages of decomposition occurred in similar temperature ranges, which indicates that crosslinking with vanillin does not influence the thermal properties of these biodegradable materials significantly.

The mass change results collected in Table 3 show that the moisture content of carboxymethyl starch is lower than that for chitosan. This may be explained by the higher hydrogen association of chitosan chains. The data presented in Figure 4 and in Table 3 clearly show that the thermal stability parameters, in particular, the T_50%_ values, depend on the amount of vanillin contained in the superabsorbent polymers. The addition of vanillin increases the decomposition temperature of the samples and reduces the weight loss compared to the raw polymers.

Chitosan and carboxymethyl starch materials with varying vanillin concentrations had a higher decomposition temperature, with one major weight loss at around 288 °C, compared to pure polymers, which showed several stages of weight loss (Appendix A). However, the most thermally stable turned out to be the superabsorbent without the addition of vanillin (Table 3).

The obtained results confirm the good compatibility between carboxymethyl starch and chitosan, showing a strong interaction between the chains of these two polymers.

### 2.4. Scanning Electron Microscopy

The microstructure of a polymer made of carboxymethyl starch and chitosan cross-linked with varying amounts of vanillin cross-linker was observed by scanning electron microscopy (Figure 5). Carboxymethyl starch (CMS) is present in granules with sharp edges and a compact, rough structure that is responsible for absorbing water. They can be compared with the structure of native corn starch, but CMS has holes and cracks that increase the surface area and absorption capacity [25]. When analyzing the SEM images of the resulting CMS polymers, an altered structure of the material with a large number of cracks, holes, and channels was observed. This fact confirms the assumption that the crosslinking described in Section 3.1 significantly changes the morphology of the reaction products. The magnification (Figure 5b,d,f,h,j,l) illustrates the inhomogeneous distribution of small pores, which can contribute to increasing the diffusion of fluids into the interior of the particle and demonstrates the formation of a continuous and stable three-dimensional lattice structure. The structure of vanillin is completely crystalline [26]. The resulting polymeric materials do not show significant crystal structures, indicating that vanillin is evenly distributed in the polymer matrix and does not separate from it. Larger pores may be the result of a low cross-link density, while densely cross-linked pores are selectively aggregated and assume a granular structure. The reasons for this may be the different degree of penetration and the degree of affinity of the liquid to the polymer network. Smaller pores can also result from an increase in the flexibility of the polymer, which is dependent on the degree of cross-linking. A polymer network with less cross-linking agent is more flexible and allows more liquid to be adsorbed because the pores are able to expand. Figure 5a,b shows the polymerization product of interpenetrating carboxymethyl starch and chitosan chains. This structure has a smoother granule surface than those with added vanillin.

The presented SEM images show pores and spherical cracks as in most of the polymers described in the literature. There are rope cracks representing meso- and microporous systems. Such superabsorbent polymers are characterized by increased resistance to mechanical damage and can be used in agriculture. The pores with a linear structure may indicate the direction of gas escape during drying, the discussed samples were dried in a vacuum oven. All images shown in Figure 5 are available in their original size in the Appendix A.

### 2.5. Swelling Properties

#### 2.5.1. Swelling Properties in Deionized Water Depend on the Amount of Crosslinker Used

Absorption and retention properties are very important in hygiene products; therefore, these parameters were checked in the materials. Analyzing the results presented in Figure 6, it can be seen that after 60 min of immersing the samples in deionized water, the largest amounts of adsorbed liquid were displayed by the CMS(1)/Ch(1)/Van(0.10) (98 g·g^−1^) sample with the highest amount of cross-linking agent, while the smallest were displayed by CMS(1)/Ch(1) (36 g·g^−1^) without the addition of vanillin. Obviously, the longer the samples were in solution, the higher the results. However, these values vary with time depending on the cross-linking agent used. After 180 min, it can be observed that the highest values of adsorbed deionized water were recorded for CMS(1)/Ch(1)/Van(0.08) (138 g·g^−1^), while the lowest results were still obtained for the sample without the use of vanillin. The equilibrium state was recorded for the polymers after 760 min, and in this case, the results also showed a different distribution than at the beginning of the analysis. This time, the highest results were recorded for the CMS(1)/Ch(1)/Van(0.04) sample, i.e., with the lowest amount of crosslinking agent used. It should be taken into account that in some samples the outer layer initially adsorbed larger amounts of liquid, while the longer the absorption process lasted, the deeper and deeper it went into the structure of the material and only then were significant amounts of the solution absorbed.

It is well known that the equilibrium liquid content, or the degree of swelling of polymers, decreases with increasing cross-linker content, since the cross-link density of the polymer chains increases. The space between the individual networks, these channels, and spaces is reduced, thereby reducing the amount of adsorbed liquid. It should also be noted that as the concentration of vanillin increased, more hydroxyl, carbonyl, carboxyl, and amine groups in chitosan were consumed as a result of the cross-linking reaction. The hydroxyl groups react with aldehydes to form an acetal, and the amino groups, as mentioned earlier, form a Schiff base with aldehydes. The system in question is less capable of forming multiple hydrogen bonds, whereby the swelling capacity is reduced due to the resulting intermolecular and intramolecular bonds. I speculate that as the chitosan content increases, the absorption properties of the future product may improve.

#### 2.5.2. Swelling Kinetics at Room Temperature

In order to compare the results of the materials obtained so far, it was necessary to determine the swelling speed, based on which accurate conclusions can be drawn. A certain amount of polymer has been immersed in deionized water and begins to take up some liquid in proportion to the soaking time. It was assumed that the first-degree equation best describes the swelling rate *τ*:(1)$$\tau =-\frac{dQ_{t}}{dt}=k\left(Q_{eq}-Q_{t}\right)$$
where *k* is the rate constant of the first-order equation; *Q_eq_* is the amount of water at swelling equilibrium at 25 °C, and *Q_t_* is the amount of water adsorbed by the given polymer sample at time *t*. The integral equation above can be converted to:(2)$$\ln(\frac{1-Q_{t}}{Q_{eq}})=-kt$$
By ordering the plot $-\ln(\frac{1-Q_{t}}{Q_{eq}})$ as a function of t, a line with a slope of k was determined. Analyzing the graphs, it can be observed that in the beginning, the increase of the adsorbed liquid is large until the values started to increase slower and slower until the process equilibrium was achieved.

A very important issue is the osmotic pressure inside the gel, which must be overcome by a specific liquid. Therefore, at the beginning of the swelling process, when the polymer was inelastic, hard, and compact, the pressure was low and the swelling rate was high. However, the longer the material stayed in the fluid, the greater its elasticity, the higher the sponginess, and the greater the expansion of the polymer chains. As a result, the given liquid had to overcome a greater osmotic pressure, which resulted in a decrease in the absorption rate until it reached equilibrium.

On the basis of the presented results in Figure 7, it can be observed that the swelling values were the highest for the hydrogel with the lowest amount of vanillin CMS(1)/Ch(1)/Van(0.04). It can be concluded that this sample was the most porous, with larger spaces for the penetration of deionized water. Additionally, this equilibrium quantity *Q_eq_* was the highest for this sample. Depending on the amount of cross-linking agent, the liquid was absorbed in such proportions by the individual samples. By analyzing the equilibrium times for individual gels, it can be noted that the liquid was absorbed the fastest by a polymer sample of the internally permeating CMS(1)/Ch(1) networks without the use of a cross-linking agent (1706 s). Apparently, these interchain spaces were the loosest and widest there, which improved the diffusion of the liquid. According to the calculations, it was the CMS(1)/Ch(1) sample with 0.06 amount of vanillin that absorbed deionized water the slowest (2513 s), so here the pore size could be the smallest, although theoretically, it should not be the case. It may also be due to the greater amount of free COOH, and OH groups that form hydrogen bonds with water. The remaining samples achieved the following liquid absorption times: CMS(1)/Ch(1)/Van(0.04)—2203 s, CMS(1)/Ch(1)/Van(0.08)—2000 s, CMS(1)/Ch(1)/Van(0.10)—2123 s.

#### 2.5.3. Thermoresponsive Properties

The swelling of the samples at 38 °C was analyzed, which indicates the temperature of the human body, which is important when this hydrogel is used in hygiene products. It is known that hydrogels are temperature sensitive when they have an appropriate hydrophobic-hydrophilic balance [32]. Polymers characterized by the ability to associate/dissociate a hydrogen bond between polar groups are thermosensitive. As the research shows (Figure 4), the obtained polymers show a temperature-dependent swelling. To a large extent, these properties depend on the number of hydrophobic groups in the side chains of the materials.

It can be observed (Table 4) that the lowest results were obtained for the sample without the interpenetrating cross-linking agent (CMS(1)/Ch(1)). Then, an increase in the absorption properties of the samples was observed with a decrease in the vanillin content, i.e., CMS(1)/Ch(1)/Van(0.10) < CMS(1)/Ch(1)/Van(0.08) < CMS(1)/Ch(1)/Van(0.06) < CMS(1)/Ch(1)/Van(0.04), similar to samples tested at room temperature. However, the rate of liquid absorption is very important in the case of hygiene products. With increasing temperature, this speed increased and the CMS(1)/Ch(1)/Van(0.04) sample (1216 s) was the fastest, and the sample with the highest amount of CMS(1)/Ch(1)/Van(0.10) (1351 s) was the slowest. The polymer chains were flexible and the increase in temperature caused secondary interactions to be broken, creating more room for water in the gel matrix.

Comparing the obtained results at 38 °C (Figure 8) with those measured at room temperature we can observe that at the beginning of the test, the samples at elevated temperatures showed higher liquid absorption values. On the other hand, the longer the value decreased, and at the moment of reaching the swelling equilibrium, the samples obtained lower swelling results than the samples tested at room temperature (Table 5). Thermodynamic aspects can explain the situation, i.e., the swelling of the samples decreases with increasing temperature. It has been known for a long time that the mixing entropy of deionized water and hydrogels decreases as a result of the formation of cage structures and an increase in the order of water molecules. Considering the equation ΔG = ΔH − TΔS, where ΔG is the Gibbs free energy; ΔH denotes enthalpy and ΔS entropy [33]. When ΔS is negative, ΔH must also be negative due to the formation of hydrogen bonds and the recorded exothermic swelling process. This causes an unfavorable increase in the negative values of TΔS for the absorption properties and an increase in ΔG.

#### 2.5.4. Swelling Behavior in Saline Solutions

When planning the use of the resulting material in hygiene products, it is necessary to analyze the swelling response of individual samples to the action of various concentrations of sodium chloride solutions. In this research paper, this chapter is pivotal. The salt solution-sensitive polymer consists of three layers: a three-dimensional matrix of the polymer network, fluid between the polymer chains, and ionic forms. As is already well known, the swelling capacity of the hydrogel is significantly influenced by the ionic strength of the absorbed solution. Based on Figure 9 it can be observed that, as in our previous works, the swelling capacity decreased significantly with increasing NaCl concentration. This explains the effect of the added positive charge (cations) on the reduction of the anion-anion electrostatic interactions, which consequently led to a difference in the osmotic pressure between the hydrogel network and the process environment.

When comparing the obtained results of swelling in various NaCl solutions with the results obtained for samples in deionized water, a significant decrease in the value can be noticed. This is a consequence of the charge screening effect. It is known that the absorption properties are dependent on the type and valency of the cations (monovalent > divalent, etc.) and the concentration of the medium solution. The highest results were recorded for the sample with the lowest amount of cross-linking agent in the lowest concentration solution, and the higher the NaCl concentration, the lower the absorption value. A sample of vanillin added beige had lower swelling results. All samples obtained significantly lower liquid permeability results, which may be due to the ability to complex carboxyl and hydroxyl groups. The resulting intramolecular and intermolecular complexes increase the thickness of the cross-linking. Similar conclusions were drawn in our previous studies described in previous publications [34,35].

The weight gain of the absorbed liquid was measured over time to determine the effect of NaCl concentration on the materials in question. In this case, a rapid increase in the swelling of the samples at the beginning of the process was also noted, while how much longer the values took to stabilize to the equilibrium value was measured. The swelling rate parameter has the lowest values for CMS(1)/Ch(1)/Van(0.04) (Table 4) in 0.1% NaCl solution, while the sample without vanillin content obtained the highest values, which confirms the previous conclusions and dependencies. It is known that the more hydroxyl and amino groups there are in a polymer, the faster the swelling rate. Charged functional groups (e.g., –OH, –NH_2_) are responsible for changing the swelling state of hydrogel networks.

#### 2.5.5. Effect of pH on Equilibrium Swelling

Tests were carried out in which the swelling degree of the resulting hydrogels was measured in solutions with different pHs ranging from 1.0 to 13.0 at room temperature. Solutions with a specific pH were prepared by appropriately mixing 0.1 M solutions of HCl and NaOH as the absorption properties of “anionic” hydrogels are dependent on the amount of added cations to the swelling medium (it decreases). The influence of pH on the swelling capacity of the synthesized hydrogels shown in Figure 10 demonstrates that the highest swelling values are achieved for hydrogel materials in the pH range of 4.30 to 9.70. The product CMS(1)/Ch(1)/Van(0.04) adsorbed the most liquid in the solution at pH 7 (20.4965 g), where Qt was 51.55 [g·g^−1^]. 

In the acidic environment (below the acid dissociation constant (pKa) of 6.3 for the chitosan amino groups), the NH_3_ groups of chitosan were protonated, which resulted in the repulsion of the polymer chains. However, this phenomenon did not dominate the entire process, because it is the anionic groups (e.g., –COOH, –OH) derived from vanillin and CMS that may predominate in the environment, as evidenced by the obtained swelling results. It can be concluded that ionization of the carboxyl groups took place in an alkaline environment. The hydrogel sample without the cross-linker showed the lowest degree of swelling compared to the samples with added vanillin as some degree of depolymerization may occur in an alkaline medium, resulting in increased liquid uptake/swelling. Alkaline pH caused the formation of a hydrogen bond between the amino and hydroxyl groups, which could additionally lower the degree of liquid absorption. The –COOH groups dissociated to form COO^−^ which caused a partial dissociation of the hydrogen bonds. At neutral pH, the carboxyl, hydroxyl and amine groups could be ionized, and the degree of ionization in the form of some electrostatic ion pairs was the highest, as was the degree of swelling.

When analyzing the results in Table 6, it can be seen that the process speed parameter was the lowest for CMS(1)/Ch(1)/Van(0.04), while it was the highest for CMS(1)/Ch(1). In this case, the structure and the predominance of anionic functional groups were also of the greatest importance.

### 2.6. Dehydration Test

The kinetics of hydrogel dehydration at 50 °C was investigated in order to find possible applications in the field of medical devices (e.g., lenses, dressings) in the future. The experimental data was presented by plotting the measured values of the fractional water release from the hydrogel in Figure 11. All studied hydrogel samples were tested under the same conditions with the initial water content at equilibrium.

The dehydration curves of hydrogels at a given dewatering temperature show that the dewatering capacity of the samples decreases with increasing loading levels of the crosslinker. The resulting polymers containing a higher vanillin content are characterized by a longer dehydration time (Figure 11). This may be due to the higher value of the gel fraction, which results in a smaller free surface area for the transport of deionized water particles. This means that liquid transport during the dewatering process would be reduced in cross-linked hydrogels, with more tortuous lanes compared to an intermolecular permeating network (CMS(1)/Ch(1)) polymer. It is known that the degree of dehydration increases with increasing temperature. This phenomenon is attributed to the increase in the diffusion coefficient of deionized water in the hydrogel matrix and its faster contraction process. There is easier relaxation of the polymer chains at higher temperatures. The outer layers of the sample opened to release deionized water while the inner layers contracted.

### 2.7. The Pro-Ecological Aspect

#### 2.7.1. Composting Process

The composting process was completed after 12 weeks. Only a powder residue of the modular chitosan material was observed (m = 0.089 g). During the measurements, increased soil adhesion to the samples was also observed. 

Initially, a slight increase in the absorbed moisture from the environment by the analyzed materials was observed. It increased in the following days of research. Intensification takes place in the 2nd and 3rd weeks. The materials achieve the maximum degree of water/moisture absorption from the environment (this is also illustrated by the analysis of the absorption of materials). It depends on the material’s structure due to the chemical modification process to which the materials were subjected. All materials were wholly dispersed in the surrounding environment.

The material meets the requirements following the PN-EN 14995, and PN-EN-13432 standards, which constitute the decay of a research object into fragments smaller than 2 mm in less than 12 weeks. The compostability of the modular material exceeds 90%.

However, further research into the acceptability of individual ingredients is required because the polymer consists of natural parts obtained by chemical modification. Therefore, the acceptance levels should also be applied to starting materials which are synthetic parts modified in chemical processes.

#### 2.7.2. Ozone Aging

The spectroscopic analysis of samples after ozone aging, together with the determination of the carbonyl index CI is presented in Table 7, and the spectra are included in the Appendix A.

The following data analysis reveals that the CI index increases in the following series of polymer composition: CMS > Ch > CMS(1)/Ch(1) > CMS(1)/Ch(1)/Van(0.04) > CMS(1)/Ch(1)/Van(0.08), which coincides with the increased polarity of the systems and, consequently, faster progress of degradation processes in the hydrophilic environment.

Single polymers and a sample of their mixture without vanillin have larger interchain spaces than crosslinked analogs, thus these samples break down into smaller parts quickly and, completely degrade. On the other hand, crosslinked samples form tighter structures, which causes difficulty in biodegradation and the process takes more time.

#### 2.7.3. Deep and Surface Water Absorption

The time dependence of surface absorption for individual modular materials is presented in Table 8 and in the collective Figure 12.

CMS(1)/Ch(1), already in the initial measurement phase, shows by far the highest values of absorption of water vapor molecules than other materials. Here, water vapor is absorbed by the material first by the capillary mechanism of the tested polymers, and then the water vapor molecules are absorbed by the diffusion mechanism. There is an interaction between the ions of individual sample components and the polar molecules of the solvent.

On the basis of deep-water absorption included in Table 9, it can be concluded that the CMS(1)/Ch(1) and CMS(1)/Ch(1)/Van(0.04) samples have superabsorbent properties. A superabsorbent material is a test subject capable of absorbing 20 times its dry weight in water or other liquid.

The decrease in sample mass during the depth absorption test may result from exceeding the absorption point of the material and the slow dissolution of the absorber in water.

#### 2.7.4. Biological Tests—Sowing Plants from the Organization for Economic Co-Operation and Development (OECD) Group

Based on the data, the lowest increase for the samples of materials on CMS(1)/Ch(1)/Van(0.04) media can be observed, which may be due to the quality of the implanted seeds. In relation to the native sample without superabsorbers in the medium, the most intensive plant growth was observed for CMS(1)/Ch(1)/Van(0.08) in the second week of growth. However, in the next two weeks from sowing, a favorable increase in relation to the native sample N was observed for CMS(1)/Ch(1), CMS(1)/Ch(1)/Van(0.08). In the last days of the analysis, the highest growth of plant tissue in the metric diagnosis was observed for CMS(1)/Ch(1)/Van(0.08) and Chitosan.

Detailed data are shown in Table 10 and are self-explanatory.

To assess the influence of SAP materials’ biodegradation on soil toxicity, the elemental analyses of appropriate samples were conducted. In order to provide adequate context for further discussion, the maximum content of elements in polymeric materials is given in Table 11.

The analysis of the elemental composition for selected materials is presented (Table 12 and Table 13) as the following percentages of individual elements and in the graphs for the samples: CMS and CMS(1)/Ch(1)/Van(0.04), as well as the representative sample.

On the basis of selected CMS and CMS(1)/Ch(1)/Van(0.04) materials from the above data obtained from soil samples, it can be concluded that components, such as metals will migrate to the soil, e.g., Fe, Cu or Zn, while in a percentage close to the native sample. The results also indicate the presence of a given element outside the scope of the determination carried out in the elemental analysis of soil samples after the OECD planting process.

The next stage of determination concerned the analysis of the elemental composition of the above-ground part for selected samples, the so-called straw.

X-ray fluorescence spectroscopy (XRF) spectra are presented in the Appendix A.

The analysis performed on straw samples showed that the element Cu was outside the range. There was slightly more Zn and Mn for the CMS(1)/Ch(1)/Van(0.04) sample compared to the reference system.

The conducted research needs to be supplemented with further analyses of the remaining modular materials in order to obtain full characteristics.

The obtained materials meet most of the criteria for compostable and biodegradable materials. During the hydrolytic decomposition of samples, the forces maintaining the polymer chains in the cross-linking state are overcome. Excessive swelling occurs, causing cracking of the network nodes and bridges connecting the structural hydrocarbon chains of the polymer material, leading to the destabilization of permanent hydrophobic bonds, etc. These phenomena were observed both during the research composting and testing the absorbability of materials.

The infiltration of samples by surface and depth methods revealed the complex structure of the modular material consisting of an outer and an inner layer. The most favorable bonding state in materials describing the outer shell, i.e., the one with the highest cross-linking density and accordingly, the inner shell of the material characterized by the lowest density inside the superabsorber particles, are demonstrated by materials with the addition of a cross-linking agent. During ozone aging, free radicals, such as ^•^H, ^•^O^2−^ or ^•^C=C, lead to the deterioration of mechanical properties, reduction of network density, and thus to the faster dispersion and disappearance of the material in the natural environment.

The research may provide a more in-depth look at the designed materials with superabsorbent properties.

## 3. Materials and Methods

### 3.1. Materials

Corn starch (CS) ACS reagent grade (Sigma Aldrich, Poznań, Poland); commercial chitosan from crab shells with a degree of deacetylation DDA = 83.40 ± 2.40% (BioLog Heppe GmbH (Landsberg, Germany); sodium hydroxide (NaOH) (Sigma Aldrich, Poznań, Poland); hydrochloric acid (Sigma Aldrich, Poznań, Poland); monochloroacetic acid, ACS reagent grade (Sigma Aldrich, Poznań, Poland); acetic acid solution of 2% (*w*/*v*) was prepared using acetic acid (Sigma Aldrich, Poznań, Poland, purity > 99%); nitrogen gas (N_2_) technical grade; ethanol 96%_vol._ (Bioetanol AEG Ltd., Chełmża, Poland). The chemicals were used without further purification. All solutions were prepared using deionized water.

### 3.2. Synthesis of Carboxymethyl Starch (CMS)

Corn starch (4.016 g) was transferred quantitatively to a three-necked flask and 30 mL of deionized water was added. The content of the flask was heated to 90 °C and stirred for 30 min using a magnetic stirrer with a heating plate. Then the flask with the content was placed in a water bath at 60 °C and equipped with a mechanical stirrer. In the meantime, 10 mL of aqueous solutions of monochloroacetic acid (4.007 g) and sodium hydroxide (3.213 g) were prepared in beakers. After complete dissolution, both solutions were quantitatively transferred to a flask using a dropping funnel (dropping rate 1 drop/5 s). The process was carried out for 4 h. After cooling down the mixture, the product was precipitated with ethanol. The gel was allowed to precipitate completely and then cut into small pieces 5 mm × 5 mm in size. The resulting superabsorbent gel was dried in a vacuum oven at 40 °C for 24 h. Figure 13 and Figure 14 show the mechanism and appearance of the final product.

### 3.3. Preparation of a Superabsorbent Polymer Composed of CMS and Chitosan (Ch)

Deionized water along with a defined amount of carboxymethyl starch (CMS) was placed in a three-necked round bottom flask and stirred by a magnetic stirrer with heating for 2 h in a water bath at 30 °C. Chitosan (Ch) powder together with a 2% acetic acid solution was quantitatively transferred to a second round-bottom three-necked flask, and the contents were mixed with a magnetic stirrer while heating for 4 h at 35 °C. After a clear chitosan solution was obtained, the specified amount of vanillin crosslinker was added and mixing was continued for 30 min. Then freshly prepared CMS solution was added to the flask with chitosan via a dropping funnel (Table 14). The resulting mixture was stirred under the same conditions for another 3 h. After that, the mixture cooled down and the product was precipitated. The resulting natural polymer was filtered on a Buchner funnel and placed at –18 °C for 72 h. The product was then dried for 72 h at room temperature using a vacuum chamber (6 × 10^−^^2^ Torr). Figure 15 shows the mechanism of amide ling formation between chitosan and vanillin responsible for cross-linking of the material. Figure 16 and Figure 17 illustrate the synthesis apparatus and the appearance of the final product.

### 3.4. Characteristics of the Obtained Superabsorbent Polymers

#### 3.4.1. Fourier Transform Infrared Spectroscopy (FTIR)

The chemical structure of the obtained polymers was characterized by Fourier transform infrared spectroscopy (FTIR). A Bruker Vertex 70 spectrometer (Bruker Optoc GmbH, Ettlingen, Germany) was used in the wavenumber range from 4000 to 400 cm^−1^ for 16 scans with a resolution of 4 cm^−1^. The obtained FTIR spectra were normalized, and the main vibration bands were assigned to the appropriate chemical groups.

#### 3.4.2. Thermal Analysis

The thermal stability of the obtained materials was tested with the SDT 2960 Simultaneous TGA-DTA thermal analyzer (TA Instruments, Champaign, IL, USA). TGA analysis was carried out in the air atmosphere on samples of a few milligrams at a heating rate of 10 °C min^−1^ from 20 to 1000 °C. The total weight loss of the sample is equal to the peak area on the DTA curve. Recorded thermograms were analyzed using TA Universal Analysis Software.

#### 3.4.3. Scanning Electron Microscopy

The particle size and surface topography of the superabsorbent were examined using a scanning electron microscope manufactured by LEO Electron Microscopy Ltd. Cambridge, UK, model 1430 VP. The apparatus was working in SE mode under the following conditions: an accelerating voltage of 10 kV, and a working distance of about 11 mm (exact WD values are given in the figures). The surface of the granules was analyzed at three sites. The test samples were dried immediately before the analysis under vacuum at the temperature of 50 °C ± 0.1 °C for 24 h. Scanning electron microscopy was used to determine the porosity of superabsorbent polymers, shape, morphology, and size.

#### 3.4.4. Swelling Properties

Properties, such as swelling rate, degree of swelling (*Q_t_*) and equilibrium swelling (*Q_eq_*) have been determined for all SAP materials with different cross-media content. The equilibrium swelling determines the maximum weight of water per 1 g of superabsorbent. About 0.1000 g of the dried polymer sample was weighed and immersed in double-distilled water to be allowed to swell for 24 h. After filtration, the extracted gel was reweighed and the *Q_eq_* was calculated according to the following formula:(3)$$Q_{eq}\;\left[\frac{g}{g}\right]=\frac{w_{s}-w_{d}}{w_{d}}$$
where *w_d_* and *w_s_* are the weights of the dry sample and the water-swollen sample, respectively.

The degree of swelling was determined in an analogous manner, but the sample was removed from the solvent (water, NaCl solution, or buffer), after a certain time, drained from the weighed excess surface water, and then re-immersed in the solvent. The degree of swelling (water absorption) was determined according to the formula:(4)$$Q_{t}\;\left[\frac{g}{g}\right]=\frac{w_{t}-w_{d}}{w_{d}}$$
where *w_t_* is the weight of the swollen sample at a given time.

The pH swelling test was carried out by immersing about 0.1000 g of the dried superabsorbent gel in solutions at a given pH at 25 °C for 24 h. Defined pH buffer solutions were prepared by calculation with 0.1 M HCl and 0.1 M NaOH solution (controlled with a pH meter). The weight of the swollen samples was measured after drying with surface filter paper.

#### 3.4.5. Swelling Dynamics

Water retention and water absorption in superabsorbent granules of various sizes were calculated according to methods widely described in the literature [36]. Study of the absorption rate of synthesized superabsorbent polymers weighing 0.1000 ± 0.001 g were placed in previously prepared and weighed tissue bags and immersed in 250 mL of deionized water. The swelling kinetics are described by mathematical equations using empirical models. The results show the initial material swelling values below 60% weight gain of the sample. The initial swelling rate is determined by a formula derived from the Voigt viscoelasticity model connecting the spring to the damper. This equation manages the rapid transition from high initially to very slow at the end of the process [34] and looks like this:(5)$$Q_{t}\;\left[\frac{g}{g}\right]=Q_{eq}\left(1-\exp\left(-\frac{t}{\tau }\right)\right)$$
where *τ* is the so-called “speed parameter” which in the original Voigt model is called “retardation time” and as such defines the dashpot effect.

The following factors may affect the absorption properties and the swelling kinetics: temperature, pH, structural parameters of the polymer network, and the properties and type of solvents. The plot of *Q_t_* = *f (t)* is described by a simple power-law equation as follows:(6)$$Q_{t}\;\left[\frac{g}{g}\right]=\;A\;\times exp\left(-b\times {\;t}^{-\frac{1}{2}}\right)\;$$
where *A* and *b* are empirical coefficients. *A* is the amount of the equilibrium swelling (*Q_eq_*), while *b* corresponds to the diffusion rate of the solvent in the interchain spaces of the polymer. The accuracy of the model is expressed in terms of R^2^ and Fit Standard Error (FSE) along with the coefficients of the equation.

#### 3.4.6. Dehydration Tests

The dehydration test was performed at 50 °C in a vacuum oven. The dehydration kinetics were determined by the gravimetric method by weighing the samples at specified time intervals. The amount of water removed from each hydrogel was determined as follows:(7)$$W_{t}\;\left[\frac{g}{g}\right]=\frac{m_{t}-m_{d}}{m_{d}}\;$$
where *m_t_* is the mass of the sample at the time *t* of dewatering, *m_d_* = *w_d_* is the initial mass of the dry sample before the dewatering process.

#### 3.4.7. Statistical Assessment of the Data

To evaluate the statistical significance of the data all repeated measurements were collected in Microsoft Excel 2010 spreadsheets and analyzed using built-in tools. The data presented in the text are the mean values. The *t*-test was applied to check the statistical significance between groups of experimental data. A value of *p* < 0.05 has been set as a statistical significance threshold.

#### 3.4.8. The Pro-Ecological Aspect

##### Composting Process

The composting process lasted less than 12 weeks and consisted of measuring the weight and organoleptic changes of the modular materials. It was carried out in a climatic chamber with the following parameters: temperature 30 °C, WWP 80%.

Qualitative and quantitative measurements and measurements of the weight of the samples ± 0.01 g were carried out at 9, 10, 11, and 12 weeks. Preparation of test samples consisted in occluding modular materials in 140 g/m^2^ with good permeability of components during the composting process.

##### Ozone Aging

The climatic aging of the samples was carried out in an ozone chamber, the technical parameters of the process were 30 °C, 200 ppm, and 200 h. The carbonyl index CI was calculated according to the following equation:(8)$$\mathrm{Cl}=\frac{transmittance\;A}{transmittance\;B}$$
where *transmittance A* is a transmittance of the characteristic group C=C 1652 [cm^−1^], and *transmittance B* is a transmittance for the reference band 2927 [cm^−1^].

##### Deep and Surface Water Absorption

In modular materials, a test was carried out to estimate water absorption by measuring the depth and surface water absorption.

Measurements of surface water absorption are based on the contact of the modular material in contact with water or in an atmosphere of water vapor (in this case it was water vapor and measurement of the mass of samples with an accuracy of ±0.0001 g inappropriate time periods) and is most often expressed in grams of absorbed water. Plots are plotted against the amount of absorbed water as a function of the contact time with water. Deepwater absorption was determined by determining the weight percentage of the total amount of water that the polymer was able to absorb. For this purpose, the weighed amount of superabsorbent was poured over with distilled water, then the excess water is poured off above the sample surface.

A material displaying superabsorbent properties is defined as a test object capable of absorbing 20 times its dry mass of water or other liquid.

##### Biological Tests—Sowing Plants from the Organization for Economic Co-Operation and Development (OECD) Group

The substrate used for sowing breeding plants belonging to one of the OECD 208 categories was sowing on a substrate that was a universal garden soil, pH 5.5–6.5. The breeding plant used was spring barley (Hordeum vulgare). The tests were carried out on a laboratory scale in cycles of three repetitions, water was supplemented in the amount of 70–100% of the capacity. A shaded area was used during the sprouting process.

The ecotoxicity studies were conducted over a period of 3 weeks, and three trials were averaged. Superabsorbent samples were introduced into the soil under the root/grain and then covered with a layer of soil.

The ecotoxicity analysis was completed with the collection of biological material in the form of the above-ground part-dried material (the so-called straw) intended for further microelement analysis along with the soil in which the plants were sown and in which also modular materials were implemented. The aim of the elemental analysis of both the above-ground part and the soil material was to determine that the mineral components did not exceed the permissible environmental standards.

## 4. Conclusions

Hydrogels were synthesized by grafting carboxymethyl starch onto chitosan. The materials were characterized using the FTIR, TA, and SEM methods. The most likely mechanism of graft polymerization has been proposed. The FTIR results show a new peak at a wavelength in the range 1716–1776 cm^−1^. Thermal analysis proved an increase in the stability of the resulting hydrogels compared to the crude monomers of CMS and Ch. The Tg of the hydrogels appeared to decrease with increasing crosslinking. SEM photos visualized the created channels, corridors, cracks and blisters that facilitate fluid penetration, absorption and adsorption of fluids. However, most important for the design is the swelling studies. The samples absorbed the largest amounts of deionized water at room temperature. It has been proven that the amount of the used vanillin cross-linker significantly influences the absorption and descriptive properties. The swelling of the polymers was characterized by a sensitivity to a change in pH, which was tested by changing the concentration of H^+^/OH^−^ ions. The resulting materials contain amino, hydroxyl and carboxyl groups which are determined to swell. It can be assumed that the main driving force responsible for such sudden swelling changes is the ionic repulsion between the charged groups introduced into the gel matrix by external pH modulation. The swelling in NaCl solutions of various concentrations was also tested. This hydrogel network intelligently responsive to pH can be considered a good candidate for designing new drug delivery systems. In this case, the absorption properties of the materials also decreased with increasing amounts of Na^+^ and Cl^−^ ions. The kinetics of dehydration of the resulting materials at a temperature of 50° C was also investigated, which showed that the presence of vanillin in the hydrogel chain reduces the rate of water removal from the polymer and extends the process of its dehydration.

## Figures and Tables

**Figure 1 ijms-23-05386-f001:**
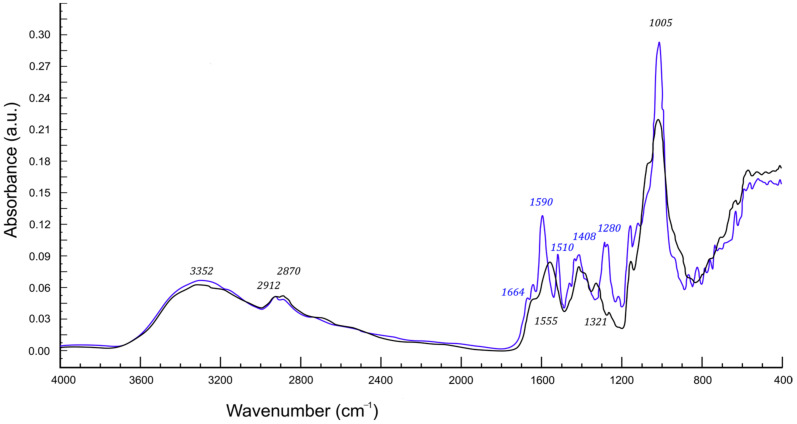
FTIR spectra of Chitosan/CMS mixture (black) vs. pure polymers’ spectra superposition (blue).

**Figure 2 ijms-23-05386-f002:**
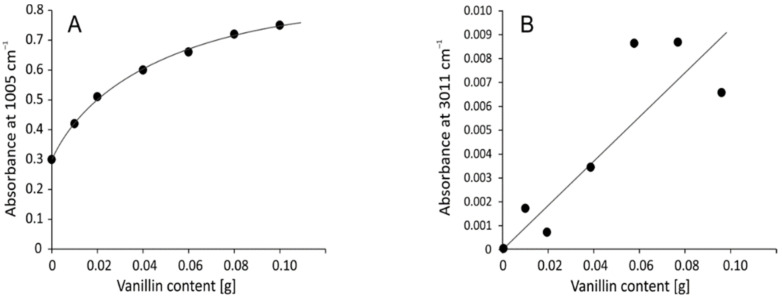
The intensity of the band at 1005 cm^−1^ (**A**) and 3011 cm^−1^ (**B**) vs. vanillin content in polymers.

**Figure 3 ijms-23-05386-f003:**
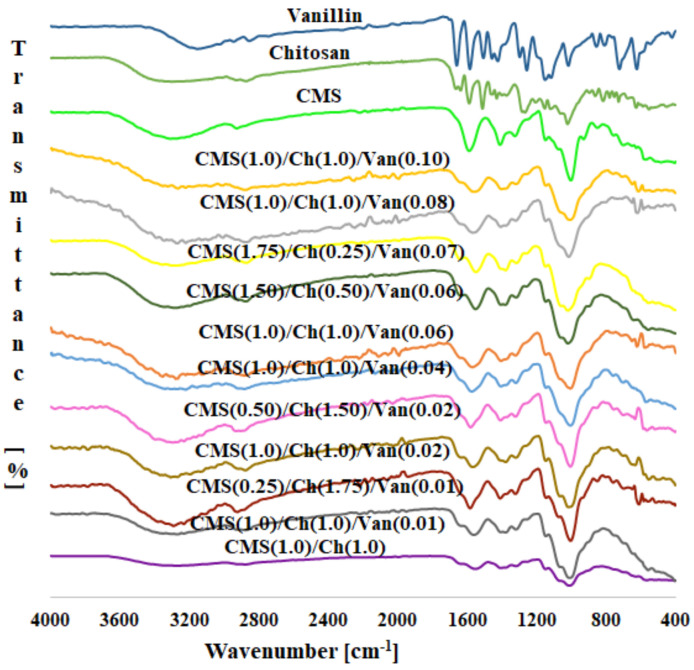
FTIR spectra of polymer samples with different amounts of crosslinking agent and raw materials used for syntheses.

**Figure 4 ijms-23-05386-f004:**
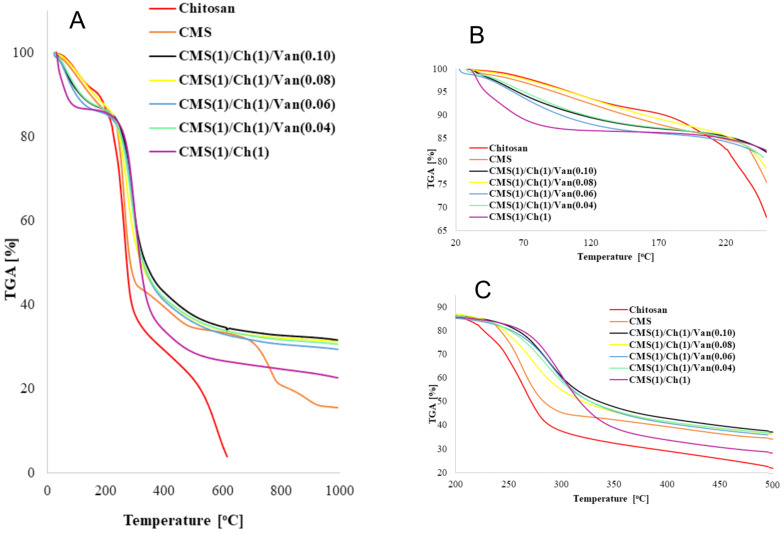
TG curves for complex superabsorbent polymers Ch and CMS in the ratio 1:1 with the addition of 0.04; 0.06; 0.08; 0.10 Van and without Van with curves for raw monomers (Ch and CMS) in various temperature ranges (**A**) the entire temperature range (0–1000 °C), (**B**) in the temperature range 20–250 °C and (**C**) in the temperature range 200–500 °C.

**Figure 5 ijms-23-05386-f005:**
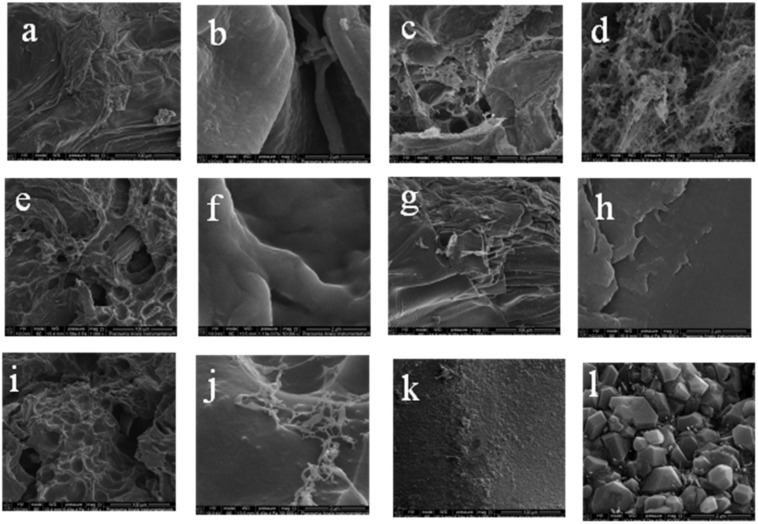
The SEM micrographs of (**a**) 1000× magnification of CMS(1)/Ch(1) without the addition of vanillin; (**b**) 50,000× magnification of CMS(1)/Ch(1) without the addition of vanillin; (**c**) 1000× magnification of CMS(1)/Ch(1)/Van(0.04); (**d**) 50,000× magnification of CMS(1)/Ch(1)/Van(0.04); (**e**) 1000× magnification of CMS(1)/Ch(1)/Van(0.06); (**f**) 50,000× magnification of CMS(1)/Ch(1)/Van(0.06); (**g**) 1000× magnification of CMS(1)/Ch(1)/Van(0.08); (**h**) 50,000× magnification of CMS(1)/Ch(1)/Van(0.08); (**i**) 1000× magnification of CMS(1)/Ch(1)/Van(0.10); (**j**) 50,000× magnification of CMS(1)/Ch(1)/Van(0.10); (**k**) 1000× magnification of CMS; (**l**) 50,000× magnification of CMS. Original full-scale images are included in the Appendix A.

**Figure 6 ijms-23-05386-f006:**
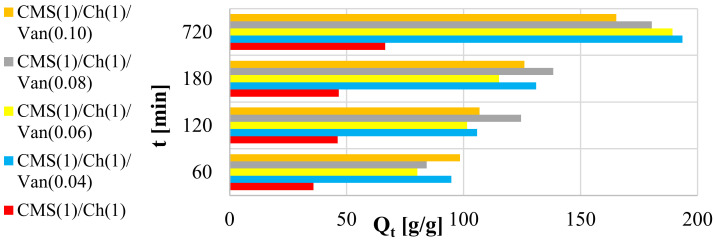
Swelling of the CMS(1)/Ch(1) polymer with a different amount of vanillin used in deionized water.

**Figure 7 ijms-23-05386-f007:**
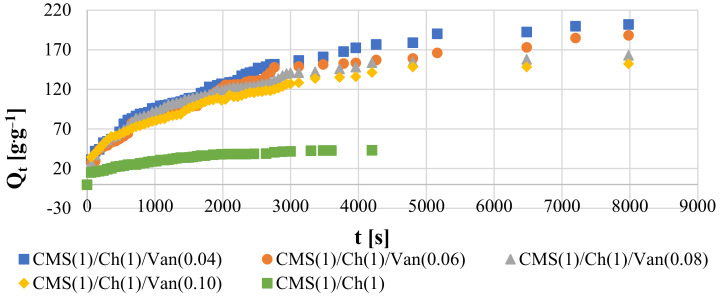
Swelling kinetic curves at 298 K.

**Figure 8 ijms-23-05386-f008:**
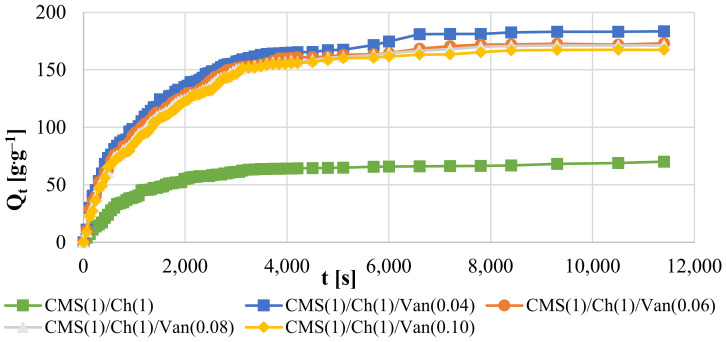
Swelling kinetic curves at 311 K.

**Figure 9 ijms-23-05386-f009:**
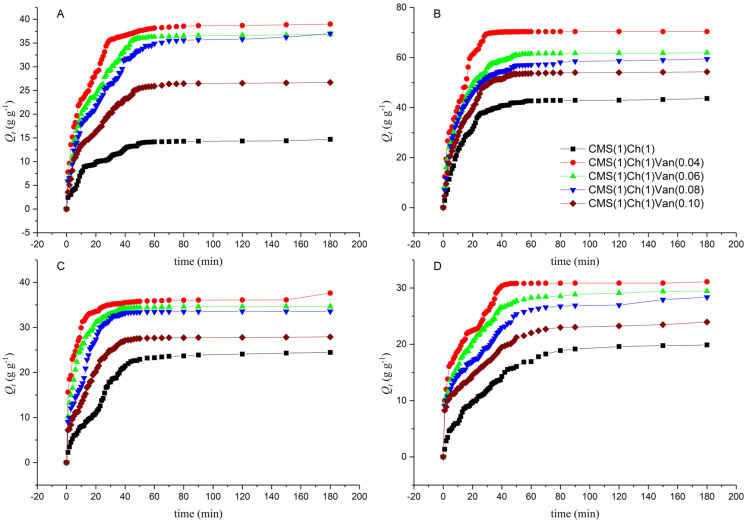
Swelling behavior of hydrogels in (**A**) 0.1% NaCl; (**B**) 0.3% NaCl; (**C**) 0.6% NaCl; (**D**) 0.9% NaCl.

**Figure 10 ijms-23-05386-f010:**
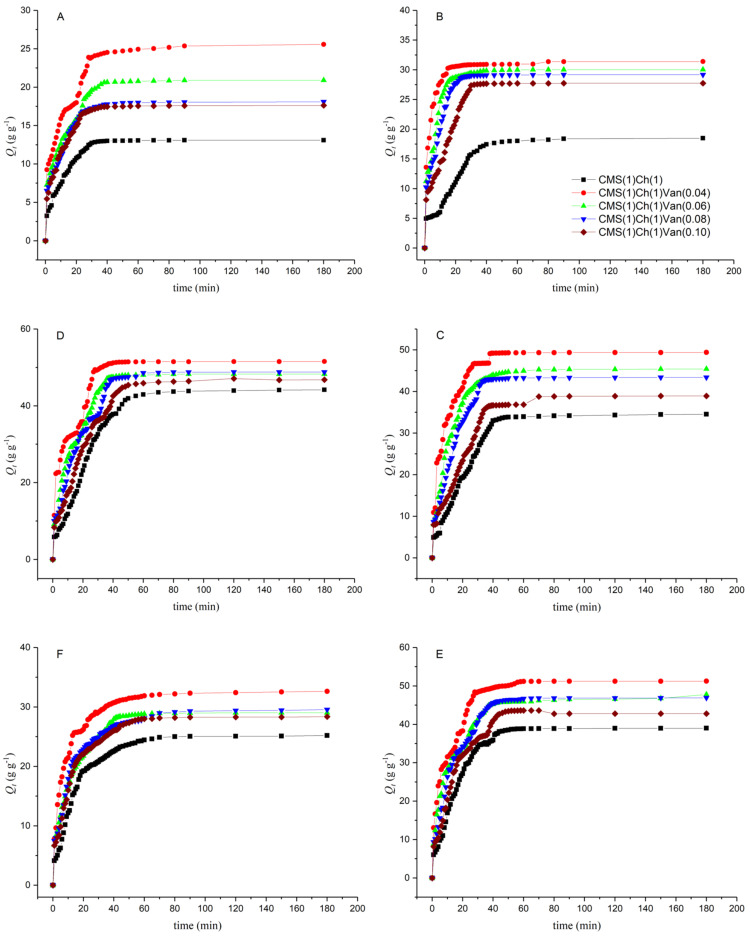
Influence of pH on the swelling capacity of synthesized hydrogels (**A**) solution with a pH of 1.48; (**B**) solution with a pH of 3.00; (**C**) solution with a pH of 4.30; (**D**) solution with a pH of 7.00; (**E**) solution with a pH of 9.70; (**F**) solution with a pH of 12.40.

**Figure 11 ijms-23-05386-f011:**
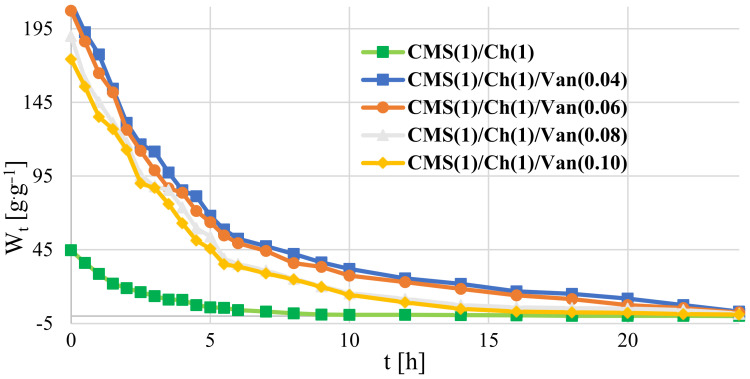
Dehydration kinetics of hydrogels at 50 °C.

**Figure 12 ijms-23-05386-f012:**
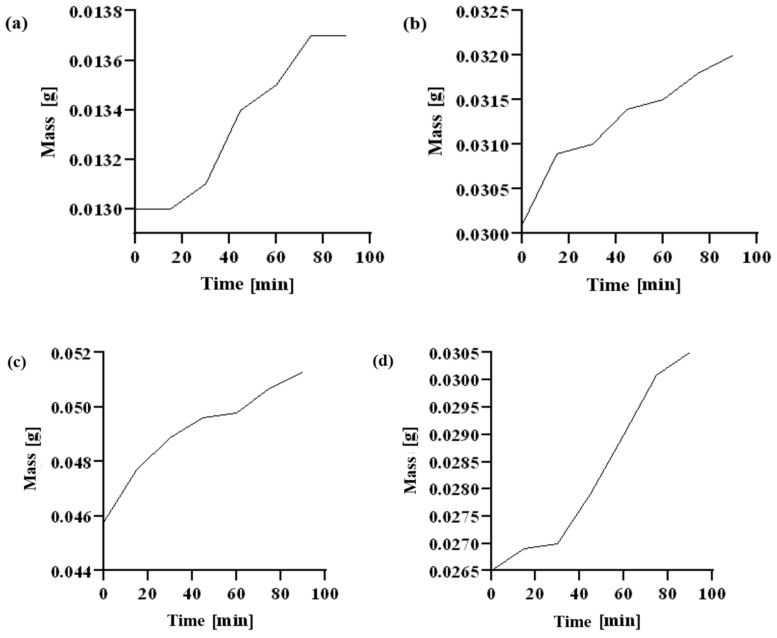
Time dependence of surface water absorption by the tested polymers: (**a**) CMS(1)/Ch(1)/Van(0.04); (**b**) CMS(1)/Ch(1)/Van(0.08); (**c**) chitosan; (**d**) CMS; (**e**) CMS(1)/Ch(1).

**Figure 13 ijms-23-05386-f013:**
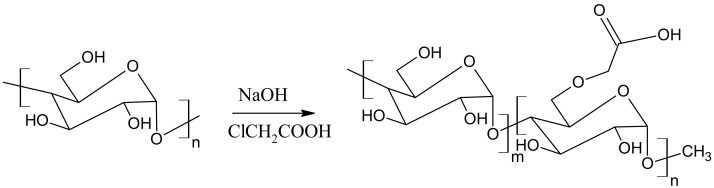
Illustration of synthetic procedure for preparation of carboxymethyl starch (CMS).

**Figure 14 ijms-23-05386-f014:**
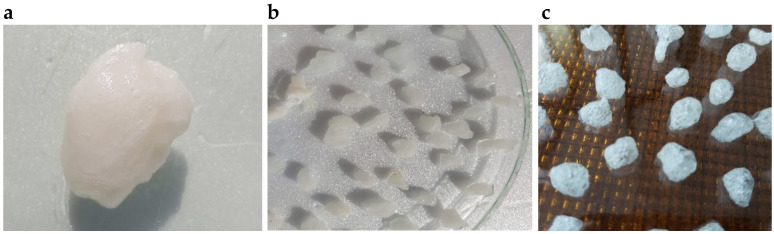
Synthesized carboxymethyl starch (**a**) immediately after precipitation with ethanol; (**b**) cut into 5 × 5 mm pieces; (**c**) dried in a vacuum oven.

**Figure 15 ijms-23-05386-f015:**
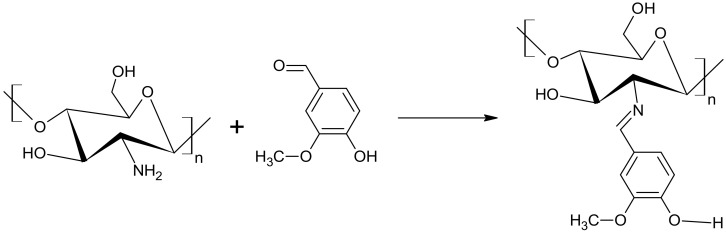
Schematic of the amide links involving vanillin (Van) carbonyl group and chitosan’s (Ch) amine group.

**Figure 16 ijms-23-05386-f016:**
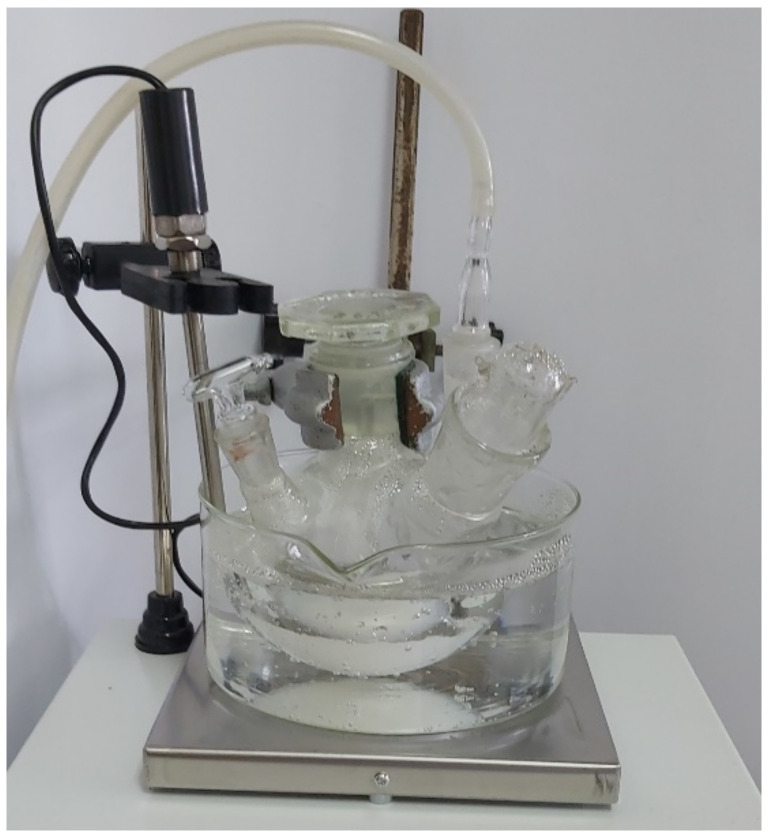
Apparatus for the synthesis of CMS and Ch in the addition of vanillin.

**Figure 17 ijms-23-05386-f017:**
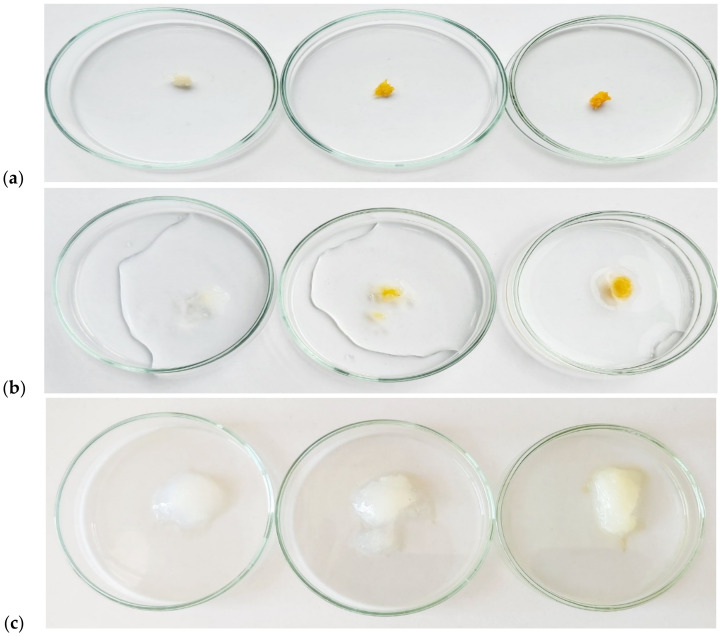
Synthesized hydrogels resulting from the polymerization of grafted carboxymethyl starch on chitosan with increasing amounts of vanillin (**a**) immediately after drying in a vacuum oven; (**b**) after adding a little deionized water; (**c**) structure and color of the hydrogel after liquid absorption.

**Table 1 ijms-23-05386-t001:** The intensity of important bands in the FTIR spectra of studied polymers.

Peak Position (cm^−1^)	Ch(1)/ CMS(1)/ Van(0)	Ch(1)/ CMS(1)/ Van(0.01)	Ch(1)/ CMS(1)/ Van(0.02)	Ch(1)/ CMS(1)/ Van(0.04)	Ch(1)/ CMS(1)/ Van(0.06)	Ch(1)/ CMS(1)/ Van(0.08)	Ch(1)/ CMS(1)/ Van(0.10)
3352	0.2503	0.1948	0.3605	0.2992	0.7009	0.6933	0.3780
3011	0.0000	0.0016	0.0006	0.0034	0.0088	0.0086	0.0062
2912	0.0291	0.0322	0.0306	0.0306	0.0321	0.0292	0.0287
1639	0.0585	0.0432	0.0581	0.0592	0.0888	0.0864	0.0752
1558	0.0973	0.0701	0.0909	0.1172	0.1674	0.1603	0.1482
1406	0.0749	0.0525	0.0583	0.0818	0.1161	0.1159	0.1205
1375	0.0721	0.0558	0.0693	0.0780	0.1108	0.1154	0.1232
1317	0.0605	0.0469	0.0579	0.0738	0.0998	0.0924	0.0967
1259	0.0187	0.0148	0.0222	0.0194	0.0321	0.0432	0.0332
1150	0.0983	0.0767	0.0844	0.1164	0.1366	0.1408	0.1397
1064	0.2399	0.2700	0.3240	0.3524	0.4180	0.4440	0.5197
1005	0.3025	0.4207	0.5099	0.6001	0.6603	0.7202	0.7501
Vanillin content	0.0000	0.0100	0.0200	0.0400	0.0600	0.0800	0.1000

**Table 2 ijms-23-05386-t002:** Correlation matrix of FTIR peaks and vanillin content.

	3352	3011	1639	1558	1406	1375	1317	1259	1150	1064	1005	
3352	1											
3011	0.774	1										
1639	0.875	0.813	1									
1558	0.760	0.872	0.951	1								
1406	0.578	0.806	0.857	0.943	1							
1375	0.573	0.779	0.843	0.902	0.966	1						
1317	0.639	0.835	0.888	0.975	0.965	0.941	1					
1259	0.759	0.758	0.836	0.784	0.758	0.827	0.725	1				
1150	0.566	0.800	0.829	0.947	0.972	0.918	0.959	0.730	1			
1064	0.391	0.686	0.594	0.702	0.762	0.862	0.792	0.680	0.748	1		
1005	0.463	0.742	0.567	0.681	0.655	0.733	0.732	0.652	0.697	0.925	1	
Van	0.399	0.726	0.606	0.726	0.806	0.885	0.803	0.724	0.799	0.983	0.911	1

**Table 3 ijms-23-05386-t003:** TG data for samples composed of chitosan and carboxymethyl starch with different vanillin content.

Name Sample	Weight Change [%] at		Temperature [°C] at 50% Weight Change	Temperature of Complete Decomposition [°C]
	120 °C (w _120 °C_)	275 °C (w _275 °C_)		
Chitosan	6.86	52.95	271.27	264.95
CMS	7.56	45.58	283.27	262.59
CMS(1)/Ch(1)/Van(0.10)	10.53	26.15	335.69	286.85
CMS(1)/Ch(1)/Van(0.08)	12.91	28.67	321.13	288.46
CMS(1)/Ch(1)/Van(0.06)	11.95	26.81	327.35	291.34
CMS(1)/Ch(1)/Van(0.04)	10.32	28.82	327.70	285.05
CMS(1)/Ch(1)	13.35	23.90	317.34	302.12

**Table 4 ijms-23-05386-t004:** Rate parameter in various concentrations of NaCl solution.

NaCl_aq_	Sample				
Concentration	CMS(1)/Ch(1)	CMS(1)/Ch(1)/ Van(0.04)	CMS(1)/Ch(1)/ Van(0.06)	CMS(1)/Ch(1)/ Van(0.08)	CMS(1)/Ch(1)/ Van(0.10)
[%]	τ [s]				
0.1	832	629	664	678	760
0.3	898	652	722	736	825
0.6	1783	685	746	765	906
0.9	1833	741	878	1049	1291

**Table 5 ijms-23-05386-t005:** Equilibrium swelling in the samples.

CMS(1)/Ch(1)	CMS(1)/Ch(1)/Van(0.04)	CMS(1)/Ch(1)/Van(0.06)	CMS(1)/Ch(1)/Van(0.08)	CMS(1)/Ch(1)/Van(0.10)
Q_eq_ [g·g^−1^]				
75.99	183.87	174.51	171.72	167.47

**Table 6 ijms-23-05386-t006:** Rate parameter in various pH buffers.

pH	Sample				
	CMS(1)/Ch(1)	CMS(1)/Ch(1)/Van(0.04)	CMS(1)/Ch(1)/Van(0.06)	CMS(1)/Ch(1)/Van(0.08)	CMS(1)/Ch(1)/Van(0.10)
	τ [s]				
1.48	214	168	174	177	185
3.00	413	47	131	178	286
4.30	512	151	252	351	470
7.00	512	122	252	351	470
9.70	364	152	166	293	347
12.40	375	142	213	246	258

**Table 7 ijms-23-05386-t007:** Detailed analysis of the CI determination for modular materials.

OZONE CHAMBER						
	Before Aging			After Aging		
	C=O	C–H	Carbonyl Index	C=O	C–H	Carbonyl Index
Chitosan	0.06087	0.05633	1.08052	0.03252	0.06346	0.51242
CMS	0.02833	0.00724	3.91533	0.07305	0.06840	1.06792
CMS(1)/Ch(1)	0.01796	0.01694	1.06011	0.10436	0.09256	1.12758
CMS(1)/Ch(1)/Van(0.04)	0.08157	0.07953	1.02573	0.10666	0.09098	1.17229
CMS(1)/Ch(1)/Van(0.08)	0.04315	0.04693	0.91940	0.08700	0.08220	1.05834

**Table 8 ijms-23-05386-t008:** Comparison of the weight results of the samples during surface water absorption.

Time [min]	Chitosan	CMS	CMS(1)/Ch(1)	CMS(1)/Ch(1)/Van(0.04)	CMS(1)/Ch(1)/Van(0.08)
0	0.0265	0.0458	0.1071	0.0130	0.0301
15	0.0269	0.0477	0.1097	0.0130	0.0309
30	0.0270	0.0489	0.1110	0.0131	0.0310
45	0.0279	0.0496	0.1119	0.0134	0.0314
60	0.0290	0.0498	0.1125	0.0135	0.0315
75	0.0301	0.0507	0.1147	0.0137	0.0318
90	0.0305	0.0513	0.1150	0.0137	0.0320

**Table 9 ijms-23-05386-t009:** Comparison of the results of the measurement of deep water absorption.

Time [h]	CMS(1)/Ch(1)	CMS(1)/Ch(1)/Van(0.04)	CMS(1)/Ch(1)/Van(0.08)
0	0.0280	0.0280	0.0353
3	0.9761	0.4289	0.3656
24	2.7959	1.0257	0.4936
48	4.2158	1.7213	0.4115
Factor	151	62	16

**Table 10 ijms-23-05386-t010:** Metric diagnosis including quantity and growth of the germinated plant.

		Name of Sample					
Time [days]		Native Sample	CMS(1)/Ch(1)/ Van(0.04)	CMS(1)/Ch(1)/ Van(0.08)	CMS(1)/Ch(1)	Chitosan	CMS
**0**	**Quantity**	**-**	**-**	**-**	**-**	**-**	**-**
	**Length [cm]**	**-**	**-**	**-**	**-**	**-**	**-**
	**Σ length [cm]**	**-**	**-**	**-**	**-**	**-**	**-**
**3**	**Quantity**	4	-	4	2	4	4
	**Length [cm]**	2.5; 3.5; 2.2; 3.0	-	3.0; 2.4; 2.0; 1.5	0.7; 0.5	2.8; 2.0; 1.2; 1.0	1.0; 1.6; 1.5; 2.0
	**Σ length [cm]**	02.9	-	2.23	0.6	1.75	1.53
**5**	**Quantity**	4	1	4	3	4	4
	**Length [cm]**	5.3; 4.3; 4.0; 4.3	little sprout	5.0; 4.0; 4.7; 3.2	1.9; 1.0; 1.2	5.8; 3.0; 4.3; 2.5	2.5; 3.8; 3.5; 4.1
	**Σ length [cm]**	4.48	little sprout	4.23	1.37	3.9	3.48
**7**	**Quantity**	4	4	4	4	5	5
	**Length [cm]**	10.2; 11.3; 11.1; 9.5	0.4; 0.9; 0.6; 1.3	12.4; 11.0; 11.2; 9.9	1.9; 5.5; 5.5; 8.0	10.4; 8.7; 11.2; 7.2; 2.0	0.4; 10.2; 9.9; 9.8; 8.4
	**Σ length [cm]**	10.53	0.8	11.13	5.23	7.9	7.74
**8**	**Quantity**	4	6	4	4	5	5
	**Length [cm]**	13.0; 14.7; 13.1; 11.7	2.7; 3.0; 3.2; 4.0; 0.6; 1.2	15.9; 14.0; 15.0; 14.6	12.0; 8.9; 9.7; 6.5	12.1; 10.5; 3.1; 7.1; 12.2	12.4; 12.0; 13.0; 13.0; 0.4
	**Σ length [cm]**	13.13	2.45	14.88	9.28	9.0	10.16
**10**	**Quantity**	4	5	4	4	5	5
	**Length [cm]**	15.5; 15.0; 15.6; 19.0	8.1; 7.6; 11.6; 2.0; 6.0	18.4; 18.1; 18.3; 20.1	15.1; 13.0; 12.2; 15.3	16.3; 15.5; 14.4; 9.0; 7.1	16.4; 16.3; 16.1; 16.9; 3.3
	**Σ length [cm]**	16.28	7.06	18.73	13.9	12.46	13.8
**15**	**Quantity**	4	6	4	4	5	5
	**Length [cm]**	15.6; 16.3; 19.5; 16.4	13.1; 10.2;13.0;14.5; 15.8;0.1	19.0; 18.5; 18.7; 24.0	15.5; 15.3; 16.3; 17.8	18.7; 15.4; 11.0; 17.4; 11.1	19.1; 6.1; 17.2; 17.6; 17.0
	**Σ length [cm]**	16.95	11.12	20.05	16.23	14.72	15.4
**19**	**Quantity**	4	6	4	4	5	5
	**Length [cm]**	15.7; 20.5; 17.4; 16.0	0.5; 15.6; 20.0; 21.8; 14.0; 10.5	19.8; 18.9; 18.8; 19.5	24.0; 19.0; 20.1; 21.0	18.9; 11.0; 15.2; 20.6; 21.2	6.4; 19.3; 19.0; 21.0; 20,.5
	**Σ length [cm]**	17.4	13.73	19.25	21.03	17.38	17.24
**27**	**Quantity**	4	6	4	4	5	5
	**Length [cm]**	24.5; 17.2; 21.7; 19.0	24.2; 11.0; 0.5; 22.9; 21.8; 22.0	23.9; 19.9; 22.3; 21.6	21.8; 26.0; 21.2; 21.0	24.1; 23.1; 20.2; 17.0; 10.2	22.2; 23.5; 18.2; 21.8; 7.0
	**Σ length [cm]**	20.6	17.07	21.93	22.5	18.92	18.54

**Table 11 ijms-23-05386-t011:** Maximum content of elements in polymeric materials.

Element	mg/kg Dry Substance	Element	mg/kg Dry Substance
Zn	150	Cr	50
Cu	50	Mo	1
Ni	25	Se	0.75
Cd	0.5	As	5
Pb	50	F	100
Hg	0.5		

**Table 12 ijms-23-05386-t012:** The analysis of the Soil elemental composition.

Native Sample							
Fe	Zn	Mn	Ti	Pd	Zr	Cu	Cr
33.411	3.017	3.095	<LOD	13.426	3.135	2.704	0.367
30.975	2.819	3.485	2.141	16.305	<LOD	0.911	<LOD
29.536	2.916	4.260	2.436	12.245	<LOD	0.405	<LOD
29.582	2.513	3.793	1.666	14.726	2.469	0.966	<LOD
**CMS**							
Fe	Zn	Mn	Ti	Pd	Zr	Cu	Cr
32.482	4.063	3.510	3.023	<LOD	<LOD	<LOD	<LOD
31.116	2.699	3.087	4.354	13.159	1.917	0.357	<LOD
29.425	3.142	3.885	4.341	<LOD	2.256	<LOD	0.593
31.47	3.275	4.061	<LOD	14.357	<LOD	2.713	0.838
32.528	2.422	2.919	3.510	15.450	<LOD	1.024	<LOD
**CMS(1)/Ch(1)/Van(0.04)**							
Fe	Zn	Mn	Ti	Pd	Zr	Cu	Cr
30.356	2.963	3.680	<LOD	16.707	2.394	2.215	1.151
32.078	2.552	3.674	<LOD	12.297	2.160	0.960	<LOD
32.100	3.613	<LOD	2.136	15.910	2.423	1.845	0.862
27.475	2.450	4.665	2.489	12.512	1.879	0.753	<LOD

**Table 13 ijms-23-05386-t013:** The analysis of the Straw elemental composition.

Zn	Mn	Ni	Cu
Native Sample			
6.736	3.442	2.04	<LOD
6.037	4.209	<LOD	2.993
6.463	3.43	1.612	<LOD
**CMS**			
6.924	3.554	1.446	<LOD
6.261	2.268	1.511	<LOD
**CMS(1)/Ch(1)/Van(0.04)**			
7.322	6.478	1.405	<LOD
7.877	6.091	1.279	<LOD
7.118	6.249	2.060	<LOD
5.612	4.800	<LOD	<LOD

**Table 14 ijms-23-05386-t014:** Explanation of polymer sample codes.

Sample Code	Description
Ch(1)/CMS(1)/Van(0)	Chitosan, carboxymethyl starch, without the addition of vanillin
Ch(1)/CMS(1)/Van(0.01)	Chitosan, carboxymethyl starch with the addition of 0.01 g of vanillin
Ch(1)/CMS(1)/Van(0.02)	Chitosan, carboxymethyl starch with the addition of 0.02 g of vanillin
Ch(1)/CMS(1)/Van(0.04)	Chitosan, carboxymethyl starch with the addition of 0.04 g of vanillin
Ch(1)/CMS(1)/Van(0.06)	Chitosan, carboxymethyl starch with the addition of 0.06 g of vanillin
Ch(1)/CMS(1)/Van(0.08)	Chitosan, carboxymethyl starch with the addition of 0.08 g of vanillin
Ch(1)/CMS(1)/Van(0.10)	Chitosan, carboxymethyl starch with the addition of 0.10 g of vanillin

## Data Availability

Not applicable.

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
