# Peer review of "Nanoarchitectonics for Biodegradable Superabsorbent Based on Carboxymethyl Starch and Chitosan Cross-Linked with Vanillin"

_ijms, 2022, doi:10.3390/ijms23105386_

Round 1
Reviewer 1 Report
- It would be better if the author explains about what the specific properties is mentioned in the page 3 line 99 of the manuscript.
- We thought that the absorbance of the materials is based on the concentration of the sample during analysis. Please determine the concentration of sample during FTIR to increase the clarity of the measurement whether it is absolute comparison or relative.
- Please consider about the margin of the paper. Because some of the figures and tables are out of position.
- Is the Figure 18 result one time measurement? No error bars founded.
Author Response
On behalf of my co-authors and myself, I want to express our thanks to the Reviewer for the valuable comments and recommendations. This helped us to improve our paper. All changes in the revised manuscript have been marked in the new version of the text. We hope that the revised version of the manuscript will find the acceptance of the Reviewer and the Editor.
Our Responses to Reviewer’s comment
Reviewer 1 (R1). It would be better if the author explains about what the specific properties is mentioned in the page 3 line 99 of the manuscript.
Authors (A). Thank you for pointing out the fault; we have amended the text according to your comment naming the most important property excluding the use of other analytical methods.
R1. We thought that the absorbance of the materials is based on the concentration of the sample during analysis. Please determine the concentration of sample during FTIR to increase the clarity of the measurement whether it is absolute comparison or relative.
A. What concerns the absorbance in Figure 7 (we guess this is what the Reviewer refers to)? We show normalized Absorbance values, which are, of course, linearly related to concentration. However, we are unable to determine the concentration of a given molecular fragment because we have no access to appropriate analytical standards. As for the concentration of the sample, we were using the ATR techniques, so in all cases, the concertation is 100%.
R1. Please consider about the margin of the paper. Because some of the figures and tables are out of position.
A. According to the Editor demands we were using special template with predefined styles margins and other layouts. So we had limited influence on the margin.
R1. Is the Figure 18 result one time measurement? No error bars founded..
A. This specific part of the experiment was done by a third party as contract research, and we do not have appropriate source data to include error bars. After consideration, we decided that Figure 18 contributed very little to the discussion. Thus, it was moved to supplementary materials.
Reviewer 2 Report
- For Figure 10 SEM images, what are they? What's the message authors would like to express? Maybe highlight e.g. arrows or pseudo color? Plus, scale bars are too small to see. What's the procedure to prepare for SEM? Does preparation affect its morphology under Vaccum on SEM?
- What's the significance of this research? Can you add a sentence in the abstract?
Author Response
On behalf of my co-authors and myself, I want to express our thanks to the Reviewer for the valuable comments and recommendations. This helped us to improve our paper. All changes in the revised manuscript have been marked in the new version of the text. We hope that the revised version of the manuscript will find the acceptance of the Reviewer and the Editor.
Responses to Reviewer’s comments
Reviewer 2 (R2). For Figure 10 SEM images, what are they? What's the message authors would like to express? Maybe highlight e.g. arrows or pseudo color? Plus, scale bars are too small to see. What's the procedure to prepare for SEM? Does preparation affect its morphology under Vacuum on SEM?
Authors (A). Thank you for pointing out the issue. The SEM images have been presented to show morphological differences between different types of synthetized materials. The images of raw materials are also included to provide a broader perspective. The specific features present in the images are mentioned and discussed in different parts of the text and, in fact, were helpful in explaining some of the data obtained by other measurements. What concerns the scale bars and analysis details they are fairly visible on full-scale images included in supplementary materials. The information was included in the new version of the Figure 10 caption. Additionally, the sample preparation procedure was included in the description method of SEM image capturing.
R2. What's the significance of this research? Can you add a sentence in the abstract?
Authors (A). Thank you for your comment on this. We have improved the Abstract including information on the significance of our research.